# Primary kidney disease modifies the effect of comorbidities on kidney replacement therapy patients' survival

Jaakko Helve [1,2]*, Mikko Haapio[2], Per-Henrik Groop[1,2,3,4], Patrik Finne[1,2]

1 Finnish Registry for Kidney Diseases, Helsinki, Finland, 2 Abdominal Center Nephrology, University of Helsinki and Helsinki University Hospital, Helsinki, Finland, 3 Folkhälsan Institute of Genetics, Folkhälsan Research Center Biomedicum Helsinki, Helsinki, Finland, 4 Department of Diabetes, Central Clinical School, Monash University, Melbourne, Australia

* Jaakko.helve@helsinki.fi

## Abstract

### Background

Comorbidities are associated with increased mortality among patients receiving long-term kidney replacement therapy (KRT). However, it is not known whether primary kidney disease modifies the effect of comorbidities on KRT patients' survival.

### Methods

An incident cohort of all patients (n = 8696) entering chronic KRT in Finland in 2000–2017 was followed until death or end of 2017. All data were obtained from the Finnish Registry for Kidney Diseases. Information on comorbidities (coronary artery disease, peripheral vascular disease, left ventricular hypertrophy, heart failure, cerebrovascular disease, malignancy, obesity, underweight, and hypertension) was collected at the start of KRT. The main outcome measure was relative risk of death according to comorbidities analyzed in six groups of primary kidney disease: type 2 diabetes, type 1 diabetes, glomerulonephritis (GN), polycystic kidney disease (PKD), nephrosclerosis, and other or unknown diagnoses. Kaplan-Meier estimates and Cox regression were used for survival analyses.

### Results

In the multivariable model, heart failure increased the risk of death threefold among PKD and GN patients, whereas in patients with other kidney diagnoses the increased risk was less than twofold. Obesity was associated with worse survival only among GN patients. Presence of three or more comorbidities increased the age- and sex-adjusted relative risk of death 4.5-fold in GN and PKD patients, but the increase was only 2.5-fold in patients in other diagnosis groups.

**Data Availability Statement:** Our data are retrieved from the patient-level data of the Finnish Registry for Kidney Diseases and sharing is restricted by the EU General Data Protection Regulation 2016/679. The data are pseudonymized but contain potentially

identifying patient information. Therefore, we do not have permission to upload our data for open access. A request to access this data can be sent to the Board of the Finnish Registry for Kidney Diseases (contact via www.muma.fi) or contact the secretary of the Finnish Registry for Kidney Diseases Heidi Niemelä (heidi.niemela@muma.fi) who will take this request to the Board of the Finnish Registry for Kidney Diseases.

**Funding:** JH was funded by Suomen Lääketieteen Säätiö (Finnish Medical Society, www.https://laaketieteensaatio.fi/) and Waldemar von Frenckells Stiftelse (http://www.foundationweb.net/frenckell/). The funders had no role in study design, data collection and analysis, decision to publish, or preparation of the manuscript.

**Competing interests:** The authors have declared that no competing interests exist.

## Conclusions

Primary kidney disease should be considered when assessing the effect of comorbidities on survival of KRT patients as it varies significantly according to type of primary kidney disease.

## Introduction

Many of the patients entering long-term kidney replacement therapy (KRT) have other medical conditions as well [1,2]. In addition, cause of kidney failure is associated with the frequency of comorbidities; for instance, KRT patients with diabetic nephropathy have more comorbidities than patients with other causes of kidney disease [3,4].

Several studies have shown that the effect of both comorbidities and of primary kidney disease is significant on KRT patients' survival [5,6]. However, Van Manen et al. [7] reported that adding comorbidities to a multivariable model on top of primary kidney disease and general patient characteristics had only a small impact on hazard ratios of death. Nevertheless, all published models predicting risk of death for patients starting dialysis assume that the effect of comorbidities on survival is constant despite primary kidney disease [8].

Knowledge on whether primary kidney disease modifies the effect of comorbidities on survival is scarce or nonexistent. This study shows how comorbidities are associated with mortality according to primary kidney disease.

## Materials and methods

All data were obtained from the Finnish Registry for Kidney Diseases. This registry has 97–99% coverage of all Finnish patients on chronic KRT since 1965. The registry is maintained by the Finnish Liver and Kidney Association and financed by the Finnish government. All patients aged 18 years or older who entered chronic KRT in an outpatient setting from 1 January 2000 to 31 December 2017 were included in the study (n = 8696). Information on age at start of KRT, sex, cause of kidney disease coded using ICD-10 codes, type of KRT (hemodialysis, peritoneal dialysis, or kidney transplantation), and time and cause of death were collected. Data on comorbidities (coronary artery disease, peripheral vascular disease, left ventricular hypertrophy, heart failure, cerebrovascular disease, and malignancy) had been collected at start of KRT as well as data on blood pressure and body mass index (BMI). The cut-off value for obesity was set at BMI $>30$ kg/m$^2$ and for underweight at BMI $<20$ kg/m$^2$.

The patients were followed from the start of KRT until death (55.1%), loss to follow-up (0.1%), moving abroad (0.2%), recovery of kidney function (1.9%), or end of follow-up on 31 December 2017 (42.7%). We divided the patients into six groups according to their primary kidney disease: type 2 diabetes, type 1 diabetes, glomerulonephritis (GN), polycystic kidney disease (PKD), nephrosclerosis, and other or unknown diagnoses. Information on comorbidities was available for 89% (left ventricular hypertrophy) to 97% (coronary artery disease) of patients. Information on all comorbidities was available for 85% of patients. Multiple imputation was used for sensitivity analyses.

The Finnish Registry for Kidney Diseases collects written informed consent from all patients for use of data for research purposes. The nephrological centers in Finland provide data on all patients at start of chronic KRT and once a year thereafter to the Finnish Registry for Kidney Diseases and these patient records are stored in the registry database. This study

was approved by the Board of the Finnish Registry for Kidney Diseases and conducted in accordance with the Data Protection Law. The patient records used here were pseudonymized, the data were encrypted, and only specified persons of the research group had access to the database. Thus, individuals cannot be identified from the results, and the study was based entirely on previously collected registry data. The dataset for this study was retrieved from the database of the Finnish Registry for Kidney Diseases in November 2018.

## Statistical methods

Between-group comparisons were performed using $\chi^2$ test for categorical variables and the Kruskal-Wallis test for continuous variables. Survival probabilities were calculated using Kaplan-Meier curves, and differences between groups were assessed using the log-rank test. Relative risks of death as a function of comorbidity were estimated using Cox proportional hazards regression with and without adjustment for potential confounding factors. Death was the event and patients were censored at the last day of follow-up or at the latest on 31 December 2017. Patients were not censored at kidney transplantation in the main analyses. Interactions between diagnosis group and comorbidities were assessed by including interaction terms in Cox regression with and without adjustment to determine whether the effect of comorbidity on survival differs according to diagnosis group. Two-sided p values lower than 0.05 were considered significant.

Statistical analyses were performed using SPSS Statistics, version 25.0.

## Results

### Baseline characteristics

Of the 8696 patients (41 793 patient-years), 4793 died during follow-up. Table 1 presents baseline characteristics, follow-up, and survival by diagnosis group. Median age at start of KRT was the lowest, 47.3 years, among patients with type 1 diabetes and the highest, 72.7 years, among patients with nephrosclerosis. Median survival time of all patients was 5.4 years,

**Table 1. Baseline characteristics, follow-up, and survival according to primary kidney disease.**

| | Type 2 diabetes | Type 1 diabetes | Glomerulo-nephritis | Polycystic kidney disease | Nephro-sclerosis | Other or unknown diagnoses | All patients | P value |
|---|---|---|---|---|---|---|---|---|
| **Number of patients (%)** | 1720 (19.8) | 1315 (15.1) | 1207 (13.9) | 826 (9.5) | 539 (6.2) | 3089 (35.5) | 8696 (100) | |
| **Kidney biopsy, %** | 13.5 | 7.6 | 86.7 | 1.8 | 27.9 | 29.1 | 28.3 | <0.001 |
| **Male gender, %** | 69.0 | 65.5 | 71.6 | 51.7 | 72.7 | 62.8 | 65.2 | <0.001 |
| **Median age at KRT start in years (25th–75th percentile)** | 67.0 (60.5–73.0) | 47.3 (39.2–55.6) | 59.7 (46.8–69.2) | 57.9 (51.1–64.8) | 71.8 (62.5–78.5) | 67.5 (56.3–75.8) | 63.2 (51.6–72.2) | <0.001 |
| **Median follow–up time in years (25th–75th percentile)** | 2.65 (1.28–4.78) | 3.87 (1.68–8.61) | 5.00 (1.97–9.56) | 5.97 (2.78–10.58) | 2.67 (1.21–5.16) | 2.95 (1.12–6.20) | 3.41 (1.41–7.02) | <0.001 |
| **Hemodialysis as the first treatment modality, %** | 83.1 | 55.6 | 68.2 | 77.7 | 76.1 | 81.1 | 75.2 | <0.001 |
| Kidney **transplantation during follow-up, N (N/100 patient-years)** | 186 (3.1) | 628 (8.7) | 609 (8.1) | 557 (9.6) | 105 (5.1) | 739 (5.6) | 2824 (6.8) | <0.001 |
| **Median survival in years (95% CI)** | 3.26 (3.08–3.43) | 7.95 (6.83–9.07) | 12.23 (10.88–3.57) | 15.41 (N/A) | 4.02 (3.53–4.51) | 4.45 (4.19–4.72) | 5.39 (5.18–5.60) | <0.001 |
| **Number of deaths (N/100 patient-years)** | 1294 (21.6) | 619 (8.6) | 445 (5.9) | 236 (4.1) | 342 (16.7) | 1857 (14.0) | 4793 (11.5) | <0.001 |
| **Cardiovascular cause of death, %** | 46.8 | 57.0 | 44.0 | 37.7 | 45.3 | 31.8 | 41.5 | <0.001 |

95% CI, 95% confidence interval.

**Table 2. Proportion of patients with comorbidities and medication at start of KRT according to primary kidney disease.**

| % | Type 2 diabetes | Type 1 diabetes | Glomerulo–nephritis | Polycystic kidney disease | Nephro–sclerosis | Other or unknown diagnoses | All patients | P value |
|---|---|---|---|---|---|---|---|---|
| **Coronary artery disease** | 41.2 | 24.5 | 14.1 | 10.7 | 39.2 | 22.3 | 25.2 | <0.001 |
| **Peripheral vascular disease** | 28.8 | 20.2 | 4.7 | 4.2 | 25.2 | 10.2 | 15.0 | <0.001 |
| **Left ventricular hypertrophy** | 48.3 | 33.2 | 26.4 | 21.9 | 51.3 | 27.2 | 33.0 | <0.001 |
| **Cerebrovascular disease** | 16.9 | 11.9 | 9.0 | 9.6 | 15.5 | 8.6 | 11.3 | <0.001 |
| **Heart failure** | 19.5 | 7.3 | 5.2 | 2.1 | 14.2 | 10.7 | 10.5 | <0.001 |
| **Malignancy** | 9.9 | 3.5 | 8.0 | 6.5 | 14.3 | 19.8 | 12.1 | <0.001 |
| **One comorbidity**[a] | 30.1 | 29.4 | 27.4 | 27.4 | 33.3 | 32.2 | 30.3 | <0.001 |
| **Two comorbidities**[a] | 24.2 | 15.4 | 10.6 | 8.9 | 21.7 | 14.9 | 16.1 | <0.001 |
| **Three or more comorbidities**[a] | 22.6 | 10.8 | 4.9 | 2.7 | 22.4 | 10.0 | 12.0 | <0.001 |
| **Obesity (BMI > 30 kg/m$^2$)** | 47.5 | 16.4 | 23.1 | 17.4 | 19.1 | 16.7 | 23.9 | <0.001 |
| **Underweight (BMI < 20 kg/m$^2$)** | 1.8 | 6.5 | 6.4 | 4.2 | 4.8 | 8.6 | 6.0 | <0.001 |
| **Systolic blood pressure > 140 mmHg** | 72.4 | 76.7 | 68.3 | 62.6 | 69.7 | 56.4 | 65.7 | <0.001 |
| **Diastolic blood pressure > 90 mmHg** | 15.5 | 31.0 | 35.5 | 31.2 | 22.6 | 20.2 | 24.3 | <0.001 |
| **Medication for hypertension** | 95.0 | 96.6 | 94.0 | 92.5 | 95.6 | 78.2 | 88.9 | <0.001 |
| **Medication for dyslipidemia** | 73.7 | 71.5 | 55.2 | 45.4 | 58.4 | 38.0 | 54.6 | <0.001 |

BMI, body mass index.

[a]Of the six comorbidities above.

ranging from 3.3 years in patients with type 2 diabetes to 15.4 years in patients with PKD. The most common cause of death was cardiovascular disease, accounting for 42% of the deceased patients. Hemodialysis was the initial treatment modality in 75% and peritoneal dialysis in 25% of patients. Of all patients, 2% started in home hemodialysis and 33% received a kidney transplant during follow-up.

More than half of the patients had at least one comorbidity (coronary artery disease, peripheral vascular disease, left ventricular hypertrophy, heart failure, cerebrovascular disease, or malignancy). Left ventricular hypertrophy (in 33%) and coronary artery disease (in 25%) were the most common comorbidities. Patients with type 2 diabetes or nephrosclerosis as the cause of kidney failure had significantly more comorbidities than other patients, and the number of comorbidities was the lowest in patients with PKD. Obesity was more common in those with type 2 diabetes than in other groups. Systolic blood pressure was elevated in two-thirds and diastolic blood pressure in one-quarter of patients despite the high prevalence of medication for hypertension (Table 2).

## Comorbidities and relative risk of death

Without adjustment, all comorbidities, obesity, and underweight were associated with increased risk of death, and only elevated blood pressure was associated with decreased risk of death regardless of whether patients were on medication for high blood pressure. The relative risk of death that was associated with each comorbidity varied significantly according to primary kidney disease. Relative risk of death associated with presence of heart failure varied the most, from 1.91 (95% CI 1.66–2.19) in patients with type 2 diabetes to 6.61 (95%

**Table 3. Multivariable model of comorbidities' effect on relative risk of death according to primary kidney disease.**

| Comorbidity, RR (95%CI) | Type 2 diabetes | Type 1 diabetes | Glomerulo-nephritis | Polycystic kidney disease | Nephro-sclerosis | Other or unknown diagnoses | All patients | Interaction P value[a] |
|---|---|---|---|---|---|---|---|---|
| Coronary artery disease | 1.18 (1.02–1.36) | 1.04 (0.85–1.26) | 1.58 (1.21–2.07) | 1.79 (1.21–2.66) | 1.09 (0.81–1.46) | 1.19 (1.04–1.36) | 1.29 (1.19–1.40) | <0.001 |
| Peripheral vascular disease | 1.80 (1.55–2.08) | 1.67 (1.32–2.11) | 1.24 (0.81–1.89) | 1.64 (0.97–2.77) | 1.47 (1.09–1.97) | 1.26 (1.07–1.50) | 1.64 (1.50–1.79) | 0.215 |
| Left ventricular hypertrophy | 1.01 (0.88–1.15) | 1.10 (0.90–1.35) | 1.17 (0.92–1.48) | 1.11 (0.79–1.56) | 1.14 (0.86–1.50) | 1.11 (0.99–1.26) | 1.13 (1.05–1.22) | 0.070 |
| Cerebrovascular disease | 1.15 (0.97–1.37) | 1.28 (0.97–1.67) | 1.52 (1.08–2.14) | 1.08 (0.68–1.72) | 1.34 (0.93–1.92) | 1.26 (1.07–1.50) | 1.24 (1.12–1.36) | 0.581 |
| Heart failure | 1.41 (1.19–1.67) | 1.81 (1.29–2.54) | 3.18 (2.10–4.83) | 3.05 (1.46–6.36) | 1.87 (1.28–2.72) | 1.85 (1.56–2.18) | 1.83 (1.65–2.03) | <0.001 |
| Malignancy | 0.92 (0.74–1.15) | 1.42 (0.91–2.21) | 1.34 (0.97–1.86) | 1.92 (1.18–3.11) | 0.72 (0.51–1.03) | 1.20 (1.06–1.36) | 1.18 (1.07–1.29) | <0.001 |
| Normal weight (BMI 20–30 kg/m$^2$) | 1 | 1 | 1 | 1 | 1 | 1 | 1 | 0.002 |
| Obesity (BMI > 30 kg/m$^2$) | 0.95 (0.83–1.08) | 1.05 (081–1.36) | 2.04 (1.59–2.61) | 0.87 (0.59–1.29) | 0.85 (0.61–1.20) | 0.98 (0.85–1.13) | 1.14 (1.05–1.22) | |
| Underweight (BMI < 20 kg/m$^2$) | 1.15 (0.70–1.88) | 2.11 (1.52–2.95) | 1.43 (0.93–2.21) | 1.45 (0.77–2.73) | 1.35 (0.74–2.46) | 1.54 (1.30–1.83) | 1.53 (1.35–1.74) | |
| Systolic blood pressure > 140 mmHg | 0.74 (0.64–0.85) | 0.83 (0.66–1.04) | 0.96 (0.76–1.23) | 1.07 (0.78–2.73) | 1.11 (0.82–1.50) | 0.89 (0.79–0.99) | 0.94 (0.87–1.01) | 0.325 |
| Diastolic blood pressure > 90 mmHg | 0.90 (0.75–1.09) | 1.00 (0.79–1.27) | 0.89 (0.67–1.18) | 0.85 (0.60–1.20) | 0.77 (0.55–1.08) | 1.03 (0.89–1.20) | 0.88 (0.80–0.96) | 0.060 |
| Age at start (per 10 years increment) | 1.45 (1.34–1.57) | 1.42 (1.28–1.57) | 2.29 (2.04–2.58) | 2.11 (1.78–2.51) | 1.87 (1.61–2.17) | 1.63 (1.56–1.72) | 1.62 (1.57–1.67) | <0.001 |
| Female sex | 1.15 (0.98–1.32) | 1.04 (0.85–1.26) | 1.02 (0.79–1.31) | 0.85 (0.63–1.15) | 0.85 (0.63–1.14) | 1.08 (0.97–1.21) | 1.00 (0.93–1.07) | 0.593 |

[a]Interaction between diagnosis group and comorbidity.

RR, relative risk of death; 95% CI, 95% confidence interval; BMI, body mass index.

Separate multivariable models were made for each diagnosis group (in columns) including all comorbidities in the table.

CI 3.57–12.22) in patients with PKD, when the risk of death was compared with patients who did not have heart failure within the same diagnosis group (S1 Table).

When adjusted for age and sex, primary kidney disease significantly modified the effect of coronary artery disease, heart failure, and malignancy on death, as all of these increased risk of death more in PKD and GN patients than in other patients (S2 Table). Obesity was associated with increased risk of death only in patients with GN. Underweight was also associated with worse prognosis, especially among patients with type 1 diabetes or nephrosclerosis.

In the multivariable model (Table 3), coronary artery disease, heart failure, and higher age had the strongest effect on worse prognosis in patients with GN or PKD. Malignancy had the most marked worsening effect on prognosis in patients with PKD. Similar to the less adjusted model, in our multivariable model the risk of death that was associated with underweight was the highest in patients with type 1 diabetes, and obesity was associated with worse prognosis only in GN patients.

Patients for whom we had all comorbidity data had better survival than patients with non-complete comorbidity data, and median survival times were 5.57 (95% CI 5.32–5.82) years and 4.63 (95% CI 4.25–5.01) years, respectively. Multiple imputation of missing comorbidity data did not change our results. When patients were censored at time of kidney transplantation, the

differences in comorbidities' association with risk of death according to primary kidney disease remained similar to results without censoring at kidney transplantation.

## Number of comorbidities and risk of death

Six comorbidities (coronary artery disease, peripheral vascular disease, left ventricular hypertrophy, heart failure, cerebrovascular disease, and malignancy) were included in the survival analyses on number of comorbidities. Increasing number of comorbidities was associated with higher risk of death in all primary kidney disease groups, but after adjustment for age and sex, the increase was more significant among patients with GN or PKD. Three or more comorbidities, compared with no comorbidity, increased risk of death 4.62-fold in patients with GN and 4.36-fold in patients with PKD, whereas relative risk varied from 2.25 to 2.63 in other primary kidney disease groups (S3 Table). Furthermore, survival time varied markedly in primary kidney disease groups according to number of comorbidities. In patients without comorbidities, median survival ranged from 4.9 years (patients with type 2 diabetes) to more than 18 years (PKD patients). However, if patients had three or more comorbidities the variation in median survival time was much smaller: from 1.5 years in GN patients to 2.9 years in PKD patients (Fig 1). This also indicates that the number of comorbidities is more relevant to the survival prognosis in patients with PKD and GN than in other diagnostic groups.

## Discussion

Our study shows that comorbidities affect KRT patients' risk of death differently depending on the type of primary kidney disease. This study included all adult patients who entered long-term KRT during 2000–2017 in Finland. A significantly greater increase in risk of death was seen in PKD and GN patients with heart failure, coronary artery disease, or malignancy than with other diagnoses, when comparing patients in the same diagnosis group with or without these comorbidities. PKD or GN patients diagnosed with heart failure before start of KRT showed a fivefold age- and sex-adjusted risk of death relative to those without this comorbidity, whereas among patients with other primary kidney diseases the increase was only twofold. PKD and GN patients had less comorbidities and better survival than other KRT patients. However, if these patients had multiple comorbidities their survival was similar to other patients with multiple comorbidities due to the greater effect of comorbidities in the survival prognosis in patients with PKD or GN than in patients with other diagnoses. To our knowledge, no previously published study has assessed this primary kidney disease-related variation in comorbidities' effect on mortality.

This nationwide study included all patients starting KRT in Finland during the 18-years study period. Data on mortality were complete, the number of patients lost to follow-up was very low, data were collected in a similar manner throughout the study period, and data on the most important comorbidities were available for a large proportion of patients. Therefore, the possibility of selection or information bias was minimal. However, a few limitations should be noted. The population in Finland is genetically quite homogeneous and almost entirely white, which may reduce the generalizability of the findings. Another potential limitation is that comorbidity data were collected mainly from pre-existing information, that is, patients were not examined separately for registry purposes. This could cause reporting bias since more severe cases were more likely reported. We also had no information on severity of comorbidities. Results of a study from the Netherlands, however, showed that adding severity grading of several comorbid conditions did not lead to improved prognostic power [9]. Although we did not have complete information on comorbidities, the data we used were significantly more comprehensive in this respect than the data used in some previous registry studies [5,10,11].

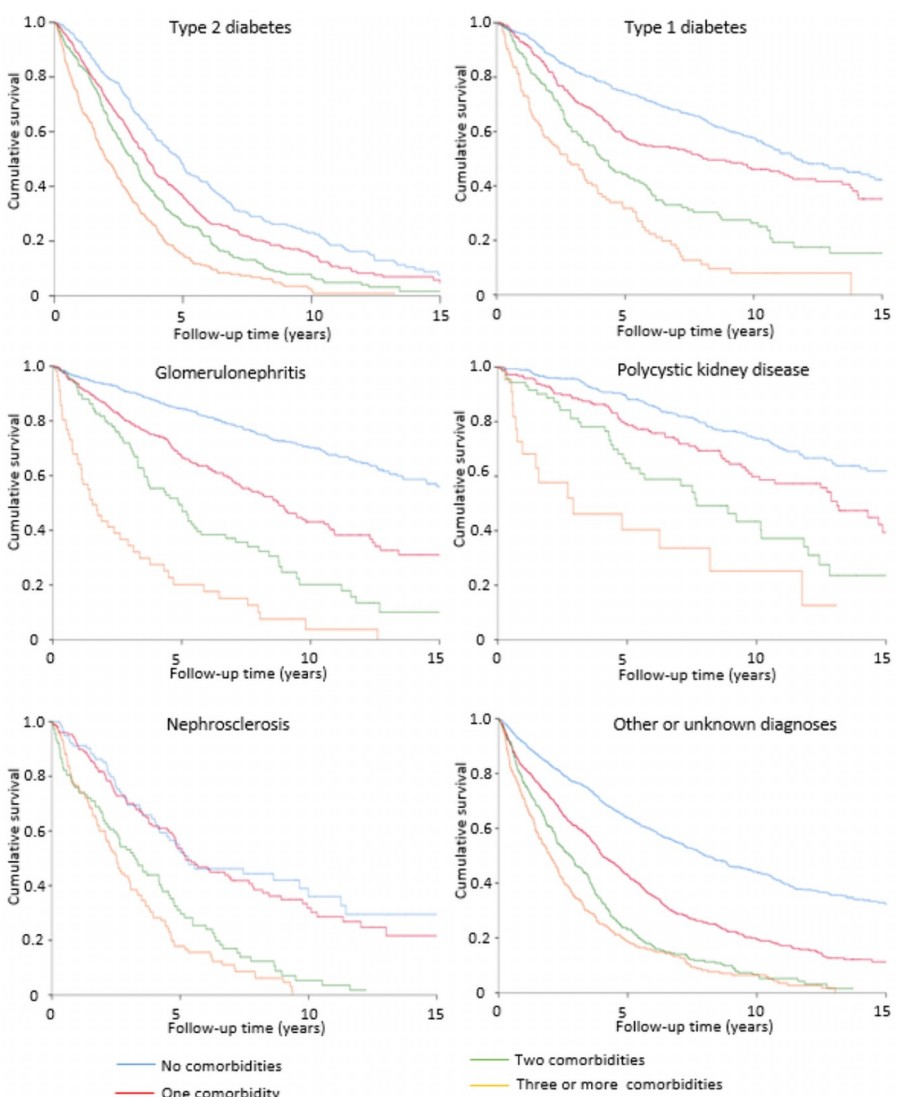

**Fig 1. Effect of the number of comorbidities on unadjusted survival probability according to primary kidney disease.**

Generally, when comparing the results of our study with results of earlier ones from other countries, it is crucial that diagnoses of primary kidney disease groups and classification of comorbid conditions are comparable. Diabetes is the leading cause of end-stage kidney disease (ESKD) in most of the countries, but incidence of ESKD caused by type 1 diabetes is higher in Finland than in other countries. The ERA-EDTA Registry, the United States Renal Data System, and previous studies have reported higher proportions of incident KRT patients with hypertension or renal vascular disease as the cause of kidney disease than we report here [5,7,12,13]. Some of the differences may emerge from differences between populations, but also classification of primary kidney disease may differ. Definitions of comorbidities and methods of data collection are likely to have an impact on reported numbers of comorbidities. Previous studies have described a higher incidence of peripheral vascular disease and heart failure at the start of KRT, whereas incidence of coronary artery disease, cerebrovascular disease, and malignancies has been similar to our study [2,4,5,7,14].

All six major comorbidities reported here have also earlier been shown to be associated with increased mortality [2,6,14–16]. Of note, van Manen et al. [7] showed that the influence of comorbidities may be less important than expected, explaining only 1.9% of the variance in mortality. Underweight has previously been established to be associated with an increased risk of death, but the association between obesity and mortality among GN patients shown here, has not been published earlier [17–20]. Although treatment of hypertension reduces mortality among patients on hemodialysis [21], hypertension has been demonstrated to have an inverse association with mortality in epidemiological studies [6,22], which is in line with our results. This may be due to an increased risk of death associated with intradialytic hypotension [23]. Taken together, our results on the entire KRT population resemble those of previous studies, and this supports the generalizability of our findings on how primary kidney disease modifies the effect of comorbidities on survival.

How could the results of our study be interpreted? Lower age in GN and PKD patients could be one explanation. If malignancy is diagnosed at a younger age, it is more likely to be more aggressive than among older patients, thus worsening the prognosis more in a young population. In addition, conservative treatment for ESKD may have been chosen more often for elderly patients who have malignancy with a poor prognosis. However, although patients with type 1 diabetes in our study were younger than GN and PKD patients, their risk of death associated with malignancy was lower. Lower risk of malignancy-related mortality in patients with diabetes than in patients with GN has been reported previously and may be caused by a higher risk of cardiovascular death in patients with diabetes as a competing factor [24]. Cardiovascular morbidity is high in patients with diabetes and in elderly patients and some of these patients may have undiagnosed coronary artery disease and heart failure. Consequently, had some of these patients been misclassified as not having these comorbidities, this would have diluted the association with comorbidity and mortality and could explain the stronger association with mortality in GN and PKD patients. In addition, higher overall risk of death in patients with diabetes and in elderly patients may reduce the additional effect of comorbidities on mortality. Furthermore, severity and type of comorbid disease at time of diagnosis and the different treatments for these comorbidities may have influenced our results.

In conclusion, comorbidities affect KRT patients' survival differently depending on the type of primary kidney disease. When assessing prognosis of patients with GN or PKD, especially cardiological problems and malignancies should be taken into account. Our findings should be considered when building prognostic models that include comorbidities, because otherwise these models are not accurate, and the estimates calculated with these models may lead to wrong conclusions for some patients. Future studies could shed more light on potential differences in severity or nature of comorbidities according to primary kidney disease. To avoid known shortcomings of an observational study like ours, a randomized study could reveal whether systematic screening and treatment of comorbidities in patients starting KRT would improve their survival.

## Supporting information

**S1 Table. Unadjusted effect of comorbidities on relative risk of death according to primary kidney disease.**
(DOCX)

**S2 Table. Age- and sex-adjusted effect of comorbidities on relative risk of death according to primary kidney disease.**
(DOCX)

**S3 Table. Age- and sex-adjusted relative risk of death according to the number of comorbidities in various groups of primary kidney disease.**
(DOCX)

## Acknowledgments

The authors acknowledge support from the Board of the Finnish Registry for Kidney Diseases and all the nephrologists and staff in all Finnish central hospitals that have reported to the Finnish Registry for Kidney Diseases: Helsinki University Central Hospital, Turku University Central Hospital, Satakunta Central Hospital, Kanta-Häme Central Hospital, Tampere University Central Hospital, Päijät-Häme Central Hospital, Kymenlaakso Central Hospital, Etelä-Karjala Central Hospital, Mikkeli Central Hospital, Itä-Savo Central Hospital, Pohjois-Karjala Central Hospital, Kuopio University Central Hospital, Keski-Suomi Central Hospital, Etelä-Pohjanmaa Central Hospital, Vaasa Central Hospital, Keski-Pohjanmaa Central Hospital, Oulu University Central Hospital, Kainuu Central Hospital, Länsi-Pohja Central Hospital, Lappi Central Hospital, and Åland Central Hospital.

## Author Contributions

**Conceptualization:** Jaakko Helve, Mikko Haapio, Per-Henrik Groop, Patrik Finne.

**Data curation:** Jaakko Helve, Patrik Finne.

**Formal analysis:** Jaakko Helve.

**Funding acquisition:** Jaakko Helve, Patrik Finne.

**Investigation:** Jaakko Helve.

**Methodology:** Jaakko Helve, Patrik Finne.

**Project administration:** Jaakko Helve.

**Resources:** Jaakko Helve, Patrik Finne.

**Software:** Jaakko Helve.

**Supervision:** Patrik Finne.

**Validation:** Jaakko Helve, Mikko Haapio.

**Visualization:** Jaakko Helve.

**Writing – original draft:** Jaakko Helve.

**Writing – review & editing:** Jaakko Helve, Mikko Haapio, Per-Henrik Groop, Patrik Finne.

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
