## [Decision Letter · Decision Letter 0]

17 Mar 2021

PONE-D-20-35936

Primary renal disease modifies the effect of comorbidities on renal replacement therapy patients’ survival

PLOS ONE

Dear Dr. Helve,

Thank you for submitting your manuscript to PLOS ONE. After careful consideration, we feel that it has merit but does not fully meet PLOS ONE’s publication criteria as it currently stands. Therefore, we invite you to submit a revised version of the manuscript that addresses the points raised during the review process.

We look forward to receiving your revised manuscript.

Kind regards,

Maria Lourdes Gonzalez Suarez, MD, PhD

Academic Editor

PLOS ONE

Additional Editor Comments (if provided):

Manuscript is of interest in the Nephrology community. It is a large cohort of patients in Finland looking at the effect of comorbidities by primary kidney disease modifying survival in patients on dialysis.

Please review and address comments made by the reviewers.

Also, please modify "renal" to "kidney" throughout the manuscript, whenever it is possible to refer to kidney disease or kidney function, as it is the new terminology recommendation by the KIDGO and to make it more patient-centered.

Journal Requirements:

2) In the ethics statement in the manuscript and in the online submission form, please provide additional information about the patient records/samples used in your retrospective study, including the date range (month and year) during which patients' medical records/samples were accessed.

3)  We note that you have indicated that data from this study are available upon request. PLOS only allows data to be available upon request if there are legal or ethical restrictions on sharing data publicly. For information on unacceptable data access restrictions, please see http://journals.plos.org/plosone/s/data-availability#loc-unacceptable-data-access-restrictions.

Reviewers' comments:

Reviewer's Responses to Questions

**Comments to the Author**

1. Is the manuscript technically sound, and do the data support the conclusions?

Reviewer #1: Yes

Reviewer #2: Partly

2. Has the statistical analysis been performed appropriately and rigorously? 

Reviewer #1: Yes

Reviewer #2: N/A

3. Have the authors made all data underlying the findings in their manuscript fully available?

Reviewer #1: Yes

Reviewer #2: Yes

4. Is the manuscript presented in an intelligible fashion and written in standard English?

Reviewer #1: No

Reviewer #2: Yes

5. Review Comments to the Author

Reviewer #1: Dear Editors,

Thank you for inviting me to review this manuscript.

This manuscript is well written with a few minor revisions. The authors were able to address the issue on whether primary renal disease modifies the effect of comorbidities on the survival of patients on chronic dialysis therapy. If published, I think this will guide clinicians make management decisions especially on how often they need to follow up with a patient. This will also help in prognostication as their data showed that patients with glomerulonephritis and polycystic kidney disease have higher risk of mortality if they have coronary artery disease or heart failure compared to other patients.

I do have some points that I would like some clarification or need some revision.

1. Their definition of overweight was a BMI > 30. Based on current criteria by the WHO, being overweight is defined as a BMI of 25.0 to <30. It might be better if they change the term overweight to obesity instead.

2. Their definition of underweight was BMI <20. Based on current criteria by the WHO, being underweight is defined as a BMI of < 18.5. However, I acknowledge that this might potentially impact their results if they adjust the cut-off. My concern is that it may misinform the readers if they used different cut-off criteria.

3. I would like to know the rationale for not including hypertension as one of the 6 comorbidities. As I compared the data from Table 2 and Table 3, it appears that they looked at patients who had elevated blood pressures and patients who were on anti-hypertensive medications. The total proportion of patients on medication was 88.9% while those with high blood pressure was 65.7% for those with elevated SBP and 24.3% for those with elevated DBP. It would be helpful for the readers to know if patients who were diagnosed with hypertension, whether controlled or not, and whether on medication or not had any significant difference in mortality.

4. The authors need to revise the language especially in the discussion part to improve readability. The specific part that needs revision is found on line 180-181, line 209-210.

Sincerely,

Carissa Dumancas, MD

Reviewer #2: The authors have described "Primary renal disease modifies the effect of comorbidities on renal replacement therapy patients’ survival". After reading the manuscript thoroughly, i suggest following modifications.

1) Authors include good cohort of RRT patients and studied retrospectively. Can authors kindly explain if its done outpatient setting only or have included patients with continuous renal replacement therapies while in hospital too.

2) Can authors mention if only hemodialysis patients were included / or patients on home therapies were included too as that could change the effect of co morbidities and primary disease on mortality.

3) Table-1 indicates that among pts with polycystic kidney disease, significant number 9.6 % underwent renal transplant as compared to type 2 DM with 3.1%. Can authors elaborate if that has any potential impact on over all mortality .

4) Table-2 indicates that most of the co morbidities including CAD, peripheral vascular disease, left ventricular hypertrophy, more than 3 Co morbidities is highest among Type 2 DM and patients with Nephrosclerosis, which in turn correlate with increased mortality. However in the results sections, authors concluded that Polycystic kidney disease, GN are associated with increase mortality. How do authors explain this variation.

5)Multiple studies mentioned uncontrolled hypertension as independent risk of mortality among ESRD patients on RRT. However the results are the study are opposite. Can authors elaborate on this

6) Appears that majority on patients in Finland are white population and hence these results could not be generalized to other parts of world with predominant black / asian population. Would be interesting to see sub group analysis on older age and race if they preset with similar findings.

6. PLOS authors have the option to publish the peer review history of their article (what does this mean?). If published, this will include your full peer review and any attached files.

Reviewer #1: No

Reviewer #2: No

---

## [Author Response · Author response to Decision Letter 0]

17 Apr 2021

The authors’ detailed response to the Additional Editors and Reviewers comments on the manuscript “Primary kidney disease modifies the effect of comorbidities on kidney replacement therapy patients’ survival”

Additional Editor Comments:

Manuscript is of interest in the Nephrology community. It is a large cohort of patients in Finland looking at the effect of comorbidities by primary kidney disease modifying survival in patients on dialysis.

Also, please modify "renal" to "kidney" throughout the manuscript, whenever it is possible to refer to kidney disease or kidney function, as it is the new terminology recommendation by the KIDGO and to make it more patient-centered.

Authors’ reply: Thank you for this notice. We have now modified the manuscript and replaced “renal” to “kidney”. 

In the ethics statement in the manuscript and in the online submission form, please provide additional information about the patient records/samples used in your retrospective study, including the date range (month and year) during which patients' medical records/samples were accessed.

Authors’ reply: We have added this more detailed information about patient records to the Materials and methods (page 5, lines 79–81 and page 6, lines 85–86).

We note that you have indicated that data from this study are available upon request. PLOS only allows data to be available upon request if there are legal or ethical restrictions on sharing data publicly. 

Authors’ reply: Our data are retrieved from the patient-level data and the sharing is restricted by the EU General Data Protection Regulation 2016/679.

Reviewer #1: 

This manuscript is well written with a few minor revisions. The authors were able to address the issue on whether primary renal disease modifies the effect of comorbidities on the survival of patients on chronic dialysis therapy. If published, I think this will guide clinicians make management decisions especially on how often they need to follow up with a patient. This will also help in prognostication as their data showed that patients with glomerulonephritis and polycystic kidney disease have higher risk of mortality if they have coronary artery disease or heart failure compared to other patients.

I do have some points that I would like some clarification or need some revision.

1. Their definition of overweight was a BMI > 30. Based on current criteria by the WHO, being overweight is defined as a BMI of 25.0 to <30. It might be better if they change the term overweight to obesity instead.

Authors’ reply: Thank you for this comment. We agree that obesity is better term for BMI > 30 and we have now changed this in the manuscript. 

2. Their definition of underweight was BMI <20. Based on current criteria by the WHO, being underweight is defined as a BMI of < 18.5. However, I acknowledge that this might potentially impact their results if they adjust the cut-off. My concern is that it may misinform the readers if they used different cut-off criteria.

Authors’ reply: We acknowledge this potential misinformation and we have now clarified this in the Methods (page 5, lines 68–69).

3. I would like to know the rationale for not including hypertension as one of the 6 comorbidities. As I compared the data from Table 2 and Table 3, it appears that they looked at patients who had elevated blood pressures and patients who were on anti-hypertensive medications. The total proportion of patients on medication was 88.9% while those with high blood pressure was 65.7% for those with elevated SBP and 24.3% for those with elevated DBP. It would be helpful for the readers to know if patients who were diagnosed with hypertension, whether controlled or not, and whether on medication or not had any significant difference in mortality.

Authors’ reply: We did not include hypertension as one of the comorbidities because it is more a risk factor for most of these other comorbidities than an individual comorbidity. The results were similar whether or not patients were on antihypertensive medication. We have added a comment on this in the Results (page 10, line 143).

4. The authors need to revise the language especially in the discussion part to improve readability. The specific part that needs revision is found on line 180-181, line 209-210.

Authors’ reply: We have now rephrased these sentences and specifically gone through the Discussion and made some changes to improve readability. We hope you find this acceptable, but we are willing to make any further changes as needed.

Reviewer #2: The authors have described "Primary renal disease modifies the effect of comorbidities on renal replacement therapy patients’ survival". After reading the manuscript thoroughly, i suggest following modifications.

1) Authors include good cohort of RRT patients and studied retrospectively. Can authors kindly explain if its done outpatient setting only or have included patients with continuous renal replacement therapies while in hospital too.

Authors’ reply: Thank you for this clarification. Only patients on chronic kidney replacement therapy and in outpatient setting were included. We have added a sentence on this issue in the Materials and methods (page 4, line 63).

2) Can authors mention if only hemodialysis patients were included / or patients on home therapies were included too as that could change the effect of co morbidities and primary disease on mortality.

Authors’ reply: Also patients on home hemodialysis and peritoneal dialysis were included. We have added this information on other treatment modalities at the initiation of kidney replacement therapy in the Results (page 7, lines 109–110).

3) Table-1 indicates that among pts with polycystic kidney disease, significant number 9.6 % underwent renal transplant as compared to type 2 DM with 3.1%. Can authors elaborate if that has any potential impact on over all mortality .

Authors’ reply: It is true that kidney transplantation is associated with better survival and may well have an impact on overall mortality but the causes (comorbidities and other factors) in patients who did not receive kidney transplant also affect mortality. Survival is better in patients with PKD than in patients with type 2 diabetes, they have fever comorbidities, and they receive more kidney transplants. However, the association between comorbidities and mortality is stronger in patients with PKD than in patients with type 2 diabetes. Therefore, the difference between the number of kidney transplants in these groups should not explain our results. In addition, the criteria for accepting patients on transplantation waitlist are the same regardless of the primary kidney disease. 

4) Table-2 indicates that most of the co morbidities including CAD, peripheral vascular disease, left ventricular hypertrophy, more than 3 Co morbidities is highest among Type 2 DM and patients with Nephrosclerosis, which in turn correlate with increased mortality. However in the results sections, authors concluded that Polycystic kidney disease, GN are associated with increase mortality. How do authors explain this variation.

Authors’ reply: As you mentioned, the patients with type 2 diabetes or nephrosclerosis have the highest number of comorbidities and the worst survival. The patients with polycystic kidney disease or glomerulonephritis have better prognosis, but in this study we show that comorbidities have a stronger worsening effect on survival among these patients. Therefore, if patients have multiple comorbidities the difference in survival is smaller between the primary kidney disease groups. 

5)Multiple studies mentioned uncontrolled hypertension as independent risk of mortality among ESRD patients on RRT. However the results are the study are opposite. Can authors elaborate on this

Authors’ reply: It is true that treatment of hypertension reduces mortality in patients on dialysis, but also other epidemiological studies than ours have shown reverse association with hypertension and mortality among RRT patients. This may be due to increased mortality associated with hypotension. We have elaborated this issue now in the Discussion and added two references (page 15, lines 234–237). 

6) Appears that majority on patients in Finland are white population and hence these results could not be generalized to other parts of world with predominant black / asian population. Would be interesting to see sub group analysis on older age and race if they preset with similar findings.

Authors’ reply: Unfortunately, we do not have the information on race. Even if we would have this information it would be difficult to present reliable results according to race because these subgroups would be too small. Also analyses according to age groups would reduce the number of patients and events too much in some diagnosis groups. For example, there were only 13 patients with type 1 diabetes over the age of 75 and 72 patients with nephrosclerosis under the age of 55. There were similar trends between diagnosis groups and number of comorbidities in different age groups as in our main results, but due to this insufficient number of patients in various subgroups, we do not think these results would be representative of this manuscript, but we can add these as needed.

---

## [Decision Letter · Decision Letter 1]

24 May 2021

PONE-D-20-35936R1

Primary kidney disease modifies the effect of comorbidities on kidney replacement therapy patients’ survival

PLOS ONE

Dear Dr. Helve,

Thank you for submitting your manuscript to PLOS ONE. After careful consideration, we feel that it has merit but does not fully meet PLOS ONE’s publication criteria as it currently stands. Therefore, we invite you to submit a revised version of the manuscript that addresses the points raised during the review process.

We look forward to receiving your revised manuscript.

Kind regards,

Maria Lourdes Gonzalez Suarez, MD, PhD

Academic Editor

PLOS ONE

Additional Editor Comments (if provided):

Thank you for submitting this revised version of your manuscript and addressing our reviewers comments. Please address new comments made by our reviewers, specifically review data in supplemental tables, and body of the manuscript were you are mention a higher risk of mortality in patients with PKD and GN when compared to the other subsets., while in the text you state that their survival rate is similar to other patients with multiple comorbidities.

Reviewers' comments:

Reviewer's Responses to Questions

**Comments to the Author**

1. If the authors have adequately addressed your comments raised in a previous round of review and you feel that this manuscript is now acceptable for publication, you may indicate that here to bypass the “Comments to the Author” section, enter your conflict of interest statement in the “Confidential to Editor” section, and submit your "Accept" recommendation.

Reviewer #1: (No Response)

Reviewer #2: All comments have been addressed

2. Is the manuscript technically sound, and do the data support the conclusions?

Reviewer #1: Partly

Reviewer #2: Yes

3. Has the statistical analysis been performed appropriately and rigorously? 

Reviewer #1: Yes

Reviewer #2: Yes

4. Have the authors made all data underlying the findings in their manuscript fully available?

Reviewer #1: Yes

Reviewer #2: Yes

5. Is the manuscript presented in an intelligible fashion and written in standard English?

Reviewer #1: No

Reviewer #2: Yes

6. Review Comments to the Author

Reviewer #1: Dear Dr. Helve,

I have reviewed the revised version of your manuscript thoroughly. Thank you for responding to my comments and clarifying the information. I am satisfied with your revisions for the cutoff scores for the obesity and underweight. I am also satisfied with your explanation why hypertension was not included as a comorbidity. It was also helpful that you mentioned that patients with higher blood pressure had lower risk of mortality which further supports the notion that intradialytic hypotension is worse for patients on KRT.

After going over the manuscript and the data, I do have some additional comments. My main take away after reading is that I agree that we must take the primary renal disease into account and not just the comorbidities of the patient when thinking about prognosis. But I think there needs to be more emphasis as to why patients with PKD or GN who have underlying heart failure, malignancy or coronary artery disease are associated with higher mortality rates compared to the other subset of patients.

• The data is clear that patients with PKD or GN have less comorbidities (Table 2) and better survival (Table 1) compared to those with diabetes or nephrosclerosis. But then both the adjusted and unadjusted data show that they have a higher risk of mortality if they have heart failure, malignancy or coronary artery disease. It might be helpful to explain why patients with PKD or GN still have better survival compared to the other subsets despite having a higher risk of mortality when they have specific comorbidities.

• Since patients with PKD or GN had higher rates of kidney transplantation and may have an impact on survival, were you able to see if the results will be affected after adjusting for kidney transplantation? If you think this is not worthwhile to look at, then maybe it will be helpful to provide a brief explanation in the manuscript as to why you think it will not impact the results of your study.

• After looking at the data on Table S3, it appears that patients who have PKD or GN with 3 or more comorbidities have a higher risk of death compared to the other subsets. But then, in the discussion, it mentioned that these patients have similar survival compared to other patients (line 198-199). Is there an explanation for this variation?

Overall, I think this version has better readability. But I think it would be helpful to go over the manuscript again for some minor grammatical revisions.

Reviewer #2: Authors have made significant improvement to raised questions. Can the authors comment in conclusions and elaborate how this study will be useful and have implication in changing the current clinical practice.

7. PLOS authors have the option to publish the peer review history of their article (what does this mean?). If published, this will include your full peer review and any attached files.

Reviewer #1: No

Reviewer #2: No

---

## [Author Response · Author response to Decision Letter 1]

30 Jun 2021

The authors’ detailed response to the Additional Editors and Reviewers comments on the manuscript “Primary kidney disease modifies the effect of comorbidities on kidney replacement therapy patients’ survival”

Additional Editor Comments:

Thank you for submitting this revised version of your manuscript and addressing our reviewers comments. Please address new comments made by our reviewers, specifically review data in supplemental tables, and body of the manuscript were you are mention a higher risk of mortality in patients with PKD and GN when compared to the other subsets., while in the text you state that their survival rate is similar to other patients with multiple comorbidities.

Authors’ reply:

In the manuscript, we show that the effect of comorbidities on the risk of death is different in different kidney disease diagnoses, and in PKD and GN patients the effect is the strongest. This results in that the survival prognosis of PKD and GN patients with multiple comorbidities is similar to that of patients in other diagnostic groups with multiple comorbidities, although the prognosis is better in PKD and GN patients without comorbidities than in patients with other kidney disease diagnoses without comorbidities. We have worked to improve the manuscript to make this message clearer. Please also see the response to these comments of the Reviewers.

Reviewer #1: 

I have reviewed the revised version of your manuscript thoroughly. Thank you for responding to my comments and clarifying the information. I am satisfied with your revisions for the cutoff scores for the obesity and underweight. I am also satisfied with your explanation why hypertension was not included as a comorbidity. It was also helpful that you mentioned that patients with higher blood pressure had lower risk of mortality which further supports the notion that intradialytic hypotension is worse for patients on KRT.

After going over the manuscript and the data, I do have some additional comments. My main take away after reading is that I agree that we must take the primary renal disease into account and not just the comorbidities of the patient when thinking about prognosis. But I think there needs to be more emphasis as to why patients with PKD or GN who have underlying heart failure, malignancy or coronary artery disease are associated with higher mortality rates compared to the other subset of patients.

• The data is clear that patients with PKD or GN have less comorbidities (Table 2) and better survival (Table 1) compared to those with diabetes or nephrosclerosis. But then both the adjusted and unadjusted data show that they have a higher risk of mortality if they have heart failure, malignancy or coronary artery disease. It might be helpful to explain why patients with PKD or GN still have better survival compared to the other subsets despite having a higher risk of mortality when they have specific comorbidities.

Authors’ reply:

Thank you for this comment because this is an essential issue for readers to understand correctly. Patients with PKD or GN without comorbidity have the lowest risk of mortality compared to the patients with other primary kidney disease without comorbidities. When PKD or GN patients have comorbidities the relative risk of death compared to PKD or GN patients without comorbidities increases more than it increases in patients with other kidney diseases when assessing the effect of comorbidities on mortality in that diagnosis group. This results a bigger difference between the survival probability of PKD or GN patients with and without comorbidities than in other primary kidney disease groups. Therefore, the survival is similar in all diagnosis groups when patients with multiple comorbidities are compared. We have clarified this issue in the Results, page 9, line 130, and page 12, line 170, and in the Discussion, page 12, lines 180 and 186.

• Since patients with PKD or GN had higher rates of kidney transplantation and may have an impact on survival, were you able to see if the results will be affected after adjusting for kidney transplantation? If you think this is not worthwhile to look at, then maybe it will be helpful to provide a brief explanation in the manuscript as to why you think it will not impact the results of your study.

Authors’ reply:

It is likely that kidney transplantation has an effect on survival and patients with PKD or GN had more often transplantation. Therefore, we made a sensitivity analysis where patients were censored at kidney transplantation and the results were similar. This is mentioned in page 11, line 154. We think that this is a better way to take an impact of kidney transplantation into account than adjustment for kidney transplantation, because in that case we would adjust the estimate of patients survival probability at the start of kidney replacement therapy with a future event (transplantation). However, we analysed how number of comorbidities associated with risk of death and multivariable model on comorbidities’ association with the risk of death with adjustment also for kidney transplantation and the results were similar, but due to the reasons mentioned above we did not add this information in the manuscript. 

• After looking at the data on Table S3, it appears that patients who have PKD or GN with 3 or more comorbidities have a higher risk of death compared to the other subsets. But then, in the discussion, it mentioned that these patients have similar survival compared to other patients (line 198-199). Is there an explanation for this variation?

Authors’ reply:

Table S3 compares the relative risk of death according to number of comorbidities within that one kidney diagnosis group. PKD patients without comorbidity have median survival time over 18 years which is higher than in other diagnosis groups, but the relative risk of death associated with number of comorbidities is greater than in PKD patients than in other diagnosis groups and therefore in PKD patients with multiple comorbidities the median survival time decreases to the same level with other patients with multiple comorbidities. We have clarified this in the manuscript, page 12, lines 170 and 186.

Overall, I think this version has better readability. But I think it would be helpful to go over the manuscript again for some minor grammatical revisions.

Authors’ reply:

The manuscript has now been revised and amended in line with the recommendations of the professional language examiner. We hope the text is clearer now.

Reviewer #2: 

Authors have made significant improvement to raised questions. Can the authors comment in conclusions and elaborate how this study will be useful and have implication in changing the current clinical practice.

Authors’ reply:

Thank you for this comment. Many prognostic models have been made to assess the survival of patients starting kidney replacement therapy. However, these models do not take into account the difference in impact of comorbidities on survival in different kidney diagnoses. If you use these models in clinical practice, it may lead to a wrong assessment in some patients and could have an influence on the decision how patient is treated. We have added a sentence on this in the Discussion, page 15, line 247.

---

## [Editor Report · Decision Letter 2]

10 Aug 2021

Primary kidney disease modifies the effect of comorbidities on kidney replacement therapy patients’ survival

PONE-D-20-35936R2

Dear Dr. Helve,

We’re pleased to inform you that your manuscript has been judged scientifically suitable for publication and will be formally accepted for publication once it meets all outstanding technical requirements.

Kind regards,

Maria Lourdes Gonzalez Suarez, MD, PhD

Academic Editor

PLOS ONE

Additional Editor Comments (optional):

Dear Jaako Helve and co-authors,

Thank you for addressing our comments and sending this revised version of your manuscript. We believe it has clarified all concerns and it has made it easier to read.

Best regards,

Maria L. Gonzalez Suarez, MD, PhD

---

## [Editor Report · Acceptance letter]

13 Aug 2021

PONE-D-20-35936R2 

Primary kidney disease modifies the effect of comorbidities on kidney replacement therapy patients’ survival 

Dear Dr. Helve:

I'm pleased to inform you that your manuscript has been deemed suitable for publication in PLOS ONE. Congratulations! Your manuscript is now with our production department. 

Kind regards, 

on behalf of

Dr. Maria Lourdes Gonzalez Suarez 

Academic Editor

PLOS ONE